

# Non-significant influence between aerobic and anaerobic sample transport materials on gut (fecal) microbiota in healthy and fat-metabolic disorder Thai adults

Naruemon Tunsakul[1], Lampet Wongsaroj[2], Kantima Janchot[3], Krit Pongpirul[3] and Naraporn Somboonna[2,4,5,6]

[1] Program in Biotechnology, Faculty of Science, Chulalongkorn University, Bangkok, Thailand
[2] Department of Microbiology, Faculty of Science, Chulalongkorn University, Bangkok, Thailand
[3] Center of Excellence in Preventive and Integrative Medicine (CE-PIM) and Department of Preventive and Social Medicine, Faculty of Medicine, Chulalongkorn University, Bangkok, Thailand
[4] Microbiome Research Unit for Probiotics in Food and Cosmetics, Chulalongkorn University, Bangkok, Thailand
[5] Omics Sciences and Bioinformatics Center, Faculty of Science, Chulalongkorn University, Bangkok, Thailand
[6] Multi-Omics for Functional Products in Food, Cosmetics and Animals Research Unit, Chulalongkorn University, Bangkok, Thailand

Corresponding author
Naraporn Somboonna, Naraporn.S@chula.ac.th

## ABSTRACT

**Background:** The appropriate sample handling for human fecal microbiota studies is essential to prevent changes in bacterial composition and quantities that could lead to misinterpretation of the data.

**Methods:** This study firstly identified the potential effect of aerobic and anaerobic fecal sample collection and transport materials on microbiota and quantitative microbiota in healthy and fat-metabolic disorder Thai adults aged 23–43 years. We employed metagenomics followed by 16S rRNA gene sequencing and 16S rRNA gene qPCR, to analyze taxonomic composition, alpha diversity, beta diversity, bacterial quantification, Pearson's correlation with clinical factors for fat-metabolic disorder, and the microbial community and species potential metabolic functions.

**Results:** Our study successfully obtained microbiota results in percent and quantitative compositions. Each sample exhibited quality sequences with a >99% Good's coverage index, and a relatively plateau rarefaction curve. Alpha diversity indices showed no statistical difference in percent and quantitative microbiota OTU richness and evenness, between aerobic and anaerobic sample transport materials. Obligate and facultative anaerobic species were analyzed and no statistical difference was observed. Supportively, the beta diversity analysis by non-metric multidimensional scale (NMDS) constructed using various beta diversity coefficients showed resembling microbiota community structures between aerobic and anaerobic sample transport groups ($P = 0.86$). On the other hand, the beta diversity could distinguish microbiota community structures between healthy and fat-metabolic disorder groups ($P = 0.02$), along with Pearson's correlated clinical parameters (*i.e.*, age, liver stiffness, GGT, BMI, and TC), the significantly associated bacterial species and their microbial metabolic functions. For example, genera such as *Ruminococcus* and *Bifidobacterium* in healthy human gut provide functions in

metabolisms of cofactors and vitamins, biosynthesis of secondary metabolites against gut pathogens, energy metabolisms, digestive system, and carbohydrate metabolism. These microbial functional characteristics were also predicted as healthy individual biomarkers by LEfSe scores. In conclusion, this study demonstrated that aerobic sample collection and transport (<48 h) did not statistically affect the microbiota and quantitative microbiota analyses in alpha and beta diversity measurements.

The study also showed that the short-term aerobic sample collection and transport still allowed fecal microbiota differentiation between healthy and fat-metabolic disorder subjects, similar to anaerobic sample collection and transport. The core microbiota were analyzed, and the findings were consistent. Moreover, the microbiota-related metabolic potentials and bacterial species biomarkers in healthy and fat-metabolic disorder were suggested with statistical bioinformatics (*i.e.*, *Bacteroides plebeius*).

# INTRODUCTION

The human intestine (gut) encompasses the complex and dynamic microbial diversity of an estimated trillion bacterial cells that are culturable and non-culturable, aerobic and anaerobic bacteria (*Human Microbiome Project (HMP) Consortium, 2012a*, *2012b*). These bacterial communities were reported to be diverse among ethnics, ages, diets, and health statuses. To date, the culture-independent microbiota study technique *via* 16S rRNA gene next-generation sequencing has been considered a reliable identification method (*Reynoso-García et al., 2022*; *Human Microbiome Project (HMP) Consortium, 2012b*).

Bacterial diversity (microbiota) in the human gut plays a vital role in maintaining health through proper fat metabolisms, prevention of gut leakage immune responses, and providing essential nutrients such as vitamins B and K, antimicrobials, and metabolites (*Reynoso-García et al., 2022*; *Valdes et al., 2018*). There are several diseases that can affect fat metabolism, cause inflammation in the bowel or autoimmune responses, trigger lupus erythematosus, or lead to cancer (*Hrncir, 2022*). Fat (or lipid) metabolism disorders occur when the body improperly processes energy from food, leading to harmful lipids deposits in organs and tissues, such as the liver, brain, and peripheral blood (*Handzlik et al., 2023*; *Yan et al., 2023*). Studies of human gut microbiota are now widely performed using fecal samples and metagenomic 16S rRNA gene high-throughput sequencing, providing culture-independent identification of bacterial diversity (*Caporaso et al., 2011*; *Dailey et al., 2019*; *Kousgaard et al., 2020*; *Human Microbiome Project (HMP) Consortium, 2012a*). Our study compared the influence of aerobic and anaerobic sample transport materials on human gut microbiota utilizing this 16S rRNA gene profiling technique and also analyzed if the microbiota differences might affect interpretation in healthy and gut disease, in which the fat-metabolic disease is presented as an example.

Numerous studies have been conducted to explore the effects of different sample collection preservatives and the duration of sample storage time on fecal samples for gut microbiome analyses. For example, a temperature of −80 °C is generally considered as the standard option for long-term storage (≥6 months), and commonly used chemicals such as 70% ethanol and a sample storage time of around 1 week have been reported as sufficient for sample preservation. Some researchers have also employed FTA cards and the OMNIgene gut kit for the same purpose (*Hsu et al., 2019*; *Ma et al., 2020*; *Song et al., 2016*; *Watson et al., 2019*). As the fecal metagenomics could be degraded, the simply general protocols recommended cold sample transport (≤4 °C) within 24–48 h after sample collection (*Gorzelak et al., 2015*; *Liang et al., 2020*; *Moossavi et al., 2019*; *Song et al., 2016*). Our study processed metagenomic extraction immediately after each sample collection and cold transport (within 24 h) to prevent this bias. Moreover, the samples were all transported by the same container material and method (closed-cap containers and by vehicle) to prevent possible microbiota diversity changes due to a bottle effect and a vehicle agitation (*Ionescu et al., 2015*; *Tihanyi-Kovács et al., 2023*). The anaerobic condition was controlled using the AnaeroPack-Anaero pack (Mitsubishi Gas Chemical, Tokyo, Japan). The effect of aerobic *vs.* anaerobic sample transport materials poses an interesting factor for local clinical sample collection settings. In local clinical settings and/or resource-constrained settings, an anaerobic sample transport material with the AnaeroPack-Anaero pack or alike is often unattainable, and the samples are collected aerobically in typical sterile closed-cap polypropylene containers without DNA preservatives (*Dore et al., 2015*; *Wesolowska-Andersen et al., 2014*). This partial aerobic condition may cause oxygen toxicity to extremely oxygen-sensitive bacteria, and thus affect fecal microbiota and quantitative microbiota analyses (*Ndongo et al., 2020*; *Taur et al., 2018*). Some bacteria, *i.e. Faecalibacterium* spp., were reported to be unable to retain cell viability for >2 min of oxygen exposure (*Duncan et al., 2002*). Limited studies have investigated the impact of aerobic and anaerobic sample transport materials without DNA preservatives on quantitative microbiota and whether this affects the ability to differentiate between healthy and metabolic-disease gut microbial diversity (*Fofanov et al., 2018*; *Jenkins et al., 2018*; *Martínez et al., 2019*). Our analyses included taxonomic composition, alpha diversity, beta diversity, bacterial quantification, between aerobic *vs.* anaerobic and between healthy *vs.* fat-metabolic disorder, and included correlation with clinical factors for fat-metabolic disorder and the microbial community and species potential metabolic functions.

## MATERIALS AND METHODS

### Participant's recruitment, fecal sample collections and metagenomic extraction

Nine healthy and eleven fat-metabolic disorder Thai participants, males and females of the age range 24–43 years, were recruited, and all methods used in this study were in accordance with the guidelines by the ethical approval. The Institutional Review Board, Faculty of Medicine, Chulalongkorn University (no. 735/61) granted the ethical approval

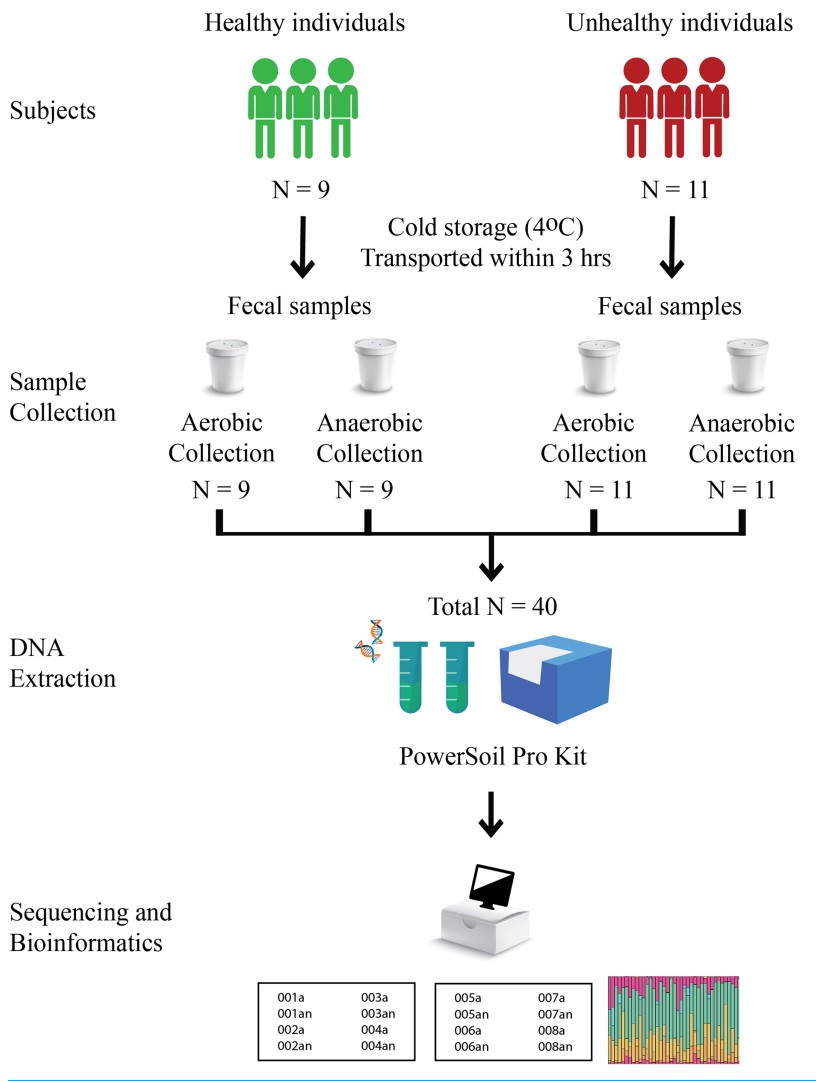

**Figure 1** **Schematic diagram of experimental design.**

for the study. Written informed consent was obtained from all participants in this study. Fecal samples of these twenty total subjects were collected in fecal containers with one aerobic and one anaerobic transport material; therefore, there were 20 aerobic transport samples and 20 anaerobic transport samples (Fig. 1). All forty samples were individually metagenomic extracted, 16S rRNA gene sequenced and qPCR for microbiota and quantitative microbiota analyses. In aspect of sample size (N), the statistically required sample size: $N = (p \,(1\text{-}p) \, z^2)/e^2$ was computed, given p at an estimated incidence between aerobic *vs.* anaerobic microbiota difference of 50%, z score of ±1.44 for 85% confidence interval, and e of 11.5% for margin of error. This yielded an N of 40 (20 aerobic and 20 anaerobic transport samples).

For aerobic transport material, the fecal container was capped, sealed and placed in a plastic bag. For anaerobic transport material, the fecal container was capped, sealed, and placed in a plastic bag with the AnaeroPack-Anaero (Mitsubishi Gas Chemical, Tokyo,

Japan) (<0.1% $O_2$ and >15% $CO_2$) (*van Horn, Warren & Baccaglini, 1997*; *Wen et al., 2021*). The samples were transported on the same day of fecal collection at a cold temperature (≤4 °C) and processed immediately within 24 h for metagenomic extraction using DNeasy PowerSoil Pro Kit (Qiagen, Hilden, Germany) following the manufacturer's instruction (*Wongsaroj et al., 2021*; *Ondee et al., 2022*). The metagenomic DNA was qualified and quantified by agarose gel electrophoresis and nanodrop spectrophotometry (A260 and A260/A280).

## 16S rRNA gene V3-V5 library preparation and MiSeq sequencing

PCR amplification of the 16S rRNA gene at the V3-V5 region was performed using the universal prokaryotic primers 342F (5′-GGRGGCAGCAGTNGGGAA-3′) and 895R (5′-TGCGDCCGTACTCCCCA-3′) with appended barcode and adaptor sequences (*Human Microbiome Project (HMP) Consortium, 2012a*; *Castelino et al., 2017*; *Wongsaroj et al., 2021*; *Dityen et al., 2022*). The 342F was used elsewhere and the 895R position was shared with the 909R. The in-silico analysis revealed that the V3-V5 primers could identify bacteria on phylum/class/order/family levels with >77% efficiency, genus 56.6% and species 21.1% (*Wang & Qian, 2009*; *Human Microbiome Project (HMP) Consortium, 2012a*; *Castelino et al., 2017*; *Johnson et al., 2019*; *Darwish et al., 2021*; *Suwarsa et al., 2021*; *Wongsaroj et al., 2021*; *Dityen et al., 2022*). Each PCR reaction comprised 1 × EmeraldAmp GT PCR Master Mix (TaKaRa, Shiga, Japan), 0.2 µM of each primer, and 50–100 ng of the genomic DNA in a total volume of 75 µL. The PCR conditions were 94 °C 3 min, and 25 cycles of 94 °C 45 s, 50 °C 1 min and 72 °C 1 min 30 s, followed by 72 °C 10 min. A minimum of two independent PCR reactions were performed and pooled to prevent PCR stochastic bias. Then, the ~640-base pair (bp) amplicon was excised from agarose gel resolution and purified using PureDireX PCR Clean-Up & Gel Extraction Kit (Bio-Helix, Keelung, Taiwan), and quantified using a Qubit 3.0 Fluorometer and Qubit dsDNA HS Assay kit (Invitrogen, Waltham, MA, USA). Finally, 180 ng of each barcoded amplicon product was pooled for sequencing using the Miseq600 platform (Illumina, San Diego, CA, USA), along with the sequencing primers and index sequence (*Caporaso et al., 2012*; *Wongsaroj et al., 2021*; *Dityen et al., 2022*; *Ondee et al., 2022*), at the Omics Sciences and Bioinformatics Center, Chulalongkorn University (Bangkok, Thailand).

## Quantification of total bacteria copy number

The 16S rRNA gene qPCR was performed to quantify total bacteria in copy unit, using universal primers 1392F (5′-CGGTGAATACGTTCYCGG-3) and 1492R (5′-GGTTACCTTGTTAC GACTT-3′), and Quantinova SYBR green PCR Master Mix (Qiagen, Hilden, Germany) in a 20 µL total volume and 1 ng metagenomic DNA (or reference DNA), as previously established (*Suzuki, Taylor & DeLong, 2000*; *Oldham & Duncan, 2012*; *Wongsaroj et al., 2021*). The qPCR thermocycling parameters were 95 °C 5 min, followed by 40 cycles of 95 °C 5 s and 60 °C 10 s. They ended with a 50–99 °C melting curve analysis to validate a single proper amplicon peak (*i.e.*, neither primer-dimer nor non-specific amplification). The reference for copy number computation was *Escherichia coli*, in which the ~120-bp 1392F-1492R amplicon fragments were cloned into
pGEM-T-Easy Vector (Promega, Wisconsin, WI, USA) and the recombinant plasmids were transformed into competent *E. coli* DH5α for expression (*Hanahan, Jessee & Bloom, 1991*). The inserted fragments were verified by colony PCR using the primers M13F (on vector) and 1492R (inserted fragment). Ten-fold serial dilutions of the extracted plasmids ($10^5$–$10^{10}$ copies/μL) were used as the reference standard curves in the bacterial copy number computation as following equation (*Smith et al., 2006*).

$$\text{Copy number per } \mu L = \frac{\text{concentration (ng/}\mu\text{L)} \times 6.023 \times 10^{23} \text{(copies/mol)}}{\text{length (bp)} \times 6.6 \times 10^{11} \text{(ng/mol)}}.$$

The qPCR experiments were performed using Rotor-GeneQ (Qiagen, Hilden, Germany). Three replicates were conducted per reaction. The bacteria copy number of each sample was quantified against the reference standard curve by Rotor-Gene Q Series Software (Qiagen, Hilden, Germany).

### Bioinformatic and statistical analyses for bacterial microbiota diversity and potential metabolisms

Raw sequences (reads) were processed following Mothur 1.39.5's standard operation procedures for MiSeq (*Schloss et al., 2009*) (https://github.com/mothur/mothur/releases/), including removal of (a) reads shorter than 100 nucleotides (nt) excluding primer and barcode sequences, (b) ambiguous bases ≥4, (c) chimera sequences, and (d) homopolymer of >7 homopolymers. The sequences were aligned with the 16S rRNA gene references and taxonomic database SILVA 13.2 (*McDonald et al., 2012*), and Greengenes 13.8 (*Quast et al., 2013*) to remove lineages of mitochondria, chloroplasts, eukaryotes, and chimera sequences. Then, the quality sequences were clustered into operational taxonomic units (OTU) with 97% nt similarity (78% for phylum, 88% order, 91% class, 93% family, 95% genus, and 97% species) based on naïve Bayesian taxonomic method with default parameters (*Wang et al., 2007*; *Schloss et al., 2009*). Samples were normalized for an equal sequencing depth (7,137 quality sequences per sample). The count of total bacteria copy number from the 16S rRNA gene qPCR data were analyzed along with the percent microbiota composition to yield the quantitative microbiota (the bacterial copy number for each individual OTU) (*Vandeputte et al., 2017a*, *2017b*; *Jian et al., 2020*; *Wongsaroj et al., 2021*). Alpha diversity including Good's coverage index (percent sequence coverage to true estimate), rarefaction curve, Chao1 richness, inverse Simpson and Shannon diversity; and beta diversity including Smith theta (Thetan), Sorenson (Sorabund), Morisita–Horn, Yue and Clayton theta (Thetayc), Bray-Curtis (BC), Jaccard (jclass), and Lennon (Lennon) coefficients, and two-dimension non-metric multidimensional scaling (NMDS), were computed using Mothur 1.39.5 (*Schloss et al., 2009*; *Schloss, 2020*). Estimates of the microbial metabolic profiles were determined by PICRUSt (Phylogenetic Investigation of Communities by Reconstruction of Unobserved States) based on the reference genome annotations in KEGG (Kyoto Encyclopedia of Genes and Genomes pathways) (*Langille et al., 2013*), and statistically compared by STAMP (Statistical Analysis of Metagenomic Profiles) (*Parks et al., 2014*). The differences in microbial metabolic profiles were further analyzed by linear discriminant analysis effect size (LEfSe) method

with pairwise Kruskal–Wallis and Wilcoxon tests to identify the microbial metabolic biomarker representing healthy and disease groups. For general statistics, non-parametric multiple $t$-tests were used and a $P$-value < 0.05 was considered significant.

### Availability of supporting data

The nucleic acid sequences in this study were deposited in the NCBI open access Sequence Read Archive database, accession number PRJNA1020208.

## RESULTS

### 16S rRNA gene sequencing results and percent microbiota compositions

The 16S rRNA gene sequencing yielded 2,365,959 total raw sequences (Table S1: aerobic sample transport 1,517,643 sequences, and anaerobic sample transport 848,316 sequences), and 1,623,517 total quality sequences (aerobic sample transport 1,062,335 sequences, and anaerobic sample transport 561,182 sequences). The average quality sequences per sample were 40,587 ± 24,139 (avg. ± SD), and the numbers of OTUs ranged 5–10 at phylum (Table 1: average 6.80 ± 1.22 OTUs), 55–93 genus, and 77–133 species levels, respectively (Table S1 and 1). The number of OTUs at phylum, genus and species levels were found approximately equal between aerobic and anaerobic sample transports (Table 1: phylum OTUs 6.55 ± 1.19 aerobic, 7.05 ± 1.23 anaerobic; genus OTUs 71.40 ± 10.45 aerobic, 72.70 ± 11.29 anaerobic; and species OTUs 101.15 ± 16.83 aerobic, 101.60 ± 15.67 anaerobic). Following the successfully high number of quality sequences, the Good's coverage (estimated percent sequence coverage to true diversity) of all samples were above 99.5% at phylum, genus and species level OTUs: avg. 100% phylum, 99.82% genus and 99.72% species (Tables 1 and S1). Once data normalization was performed of all samples, each to the same sequencing depth, the Good's coverages remained average >99% and the rarefaction curves were relative plateau (Fig. S1). The data disclosed that the further microbiota bioinformatic analyses had no bias from various quality sequencing numbers per sample.

The percent bacterial compositions at phylum, genus, and species levels across all participants were compared between aerobic $vs.$ anaerobic sample transport materials, and no statistical difference in the phylum/genus/species was found (AMOVA, $P > 0.05$) (Fig. 2). Five major phyla, ranging from Firmicutes as the top abundant (averagely, 52.03 ± 17.30%), Bacteroidetes (24.32 ± 14.11%), Proteobacteria, Actinobacteria, to Fusobacteria, were presented. The latter three phyla accounted for an average <24%. Twenty-two bacterial genera (equating 24 bacterial species OTUs), excluded <1% genus, or species, were revealed and the individual percent genus (or species) was compared between aerobic $vs.$ anaerobic sample transport materials: no statistical difference were found ($t$-test, $P > 0.05$) (Table S2). The OTU compositions indicated no statistical difference in microbiota percents and compositions at phylum, genus and species levels, between aerobic and anaerobic sample transport groups.

**Table 1 Average quality reads, OTUs and Good's coverage (%) at phylum, genus and species level.**

| Groups | Quality sequences | Phylum | | Genus | | Species | |
|---|---|---|---|---|---|---|---|
| | | OTUs | Good's coverage | OTUs | Good's coverage | OTUs | Good's coverage |
| **All** | 40,587 ± 24,139 | 6.80 ± 1.22 | 100 | 72.05 ± 10.76 | 99.82 | 101.38 ± 16.05 | 99.72 |
| **Aerobic collection** | 53,116 ± 24,211 | 6.55 ± 1.19 | 100 | 71.40 ± 10.45 | 99.83 | 101.15 ± 16.83 | 99.71 |
| **Anaerobic collection** | 28,059 ± 16,715 | 7.05 ± 1.23 | 100 | 72.70 ± 11.29 | 99.81 | 101.60 ± 15.67 | 99.72 |

**Note:**
For number of quality reads and OTUs, data were displayed in average ± SD. Multiple *t*-tests were performed for OTUs and Good's coverages between aerobic and anaerobic sample transport groups and no statistical difference was determined ($P > 0.05$).

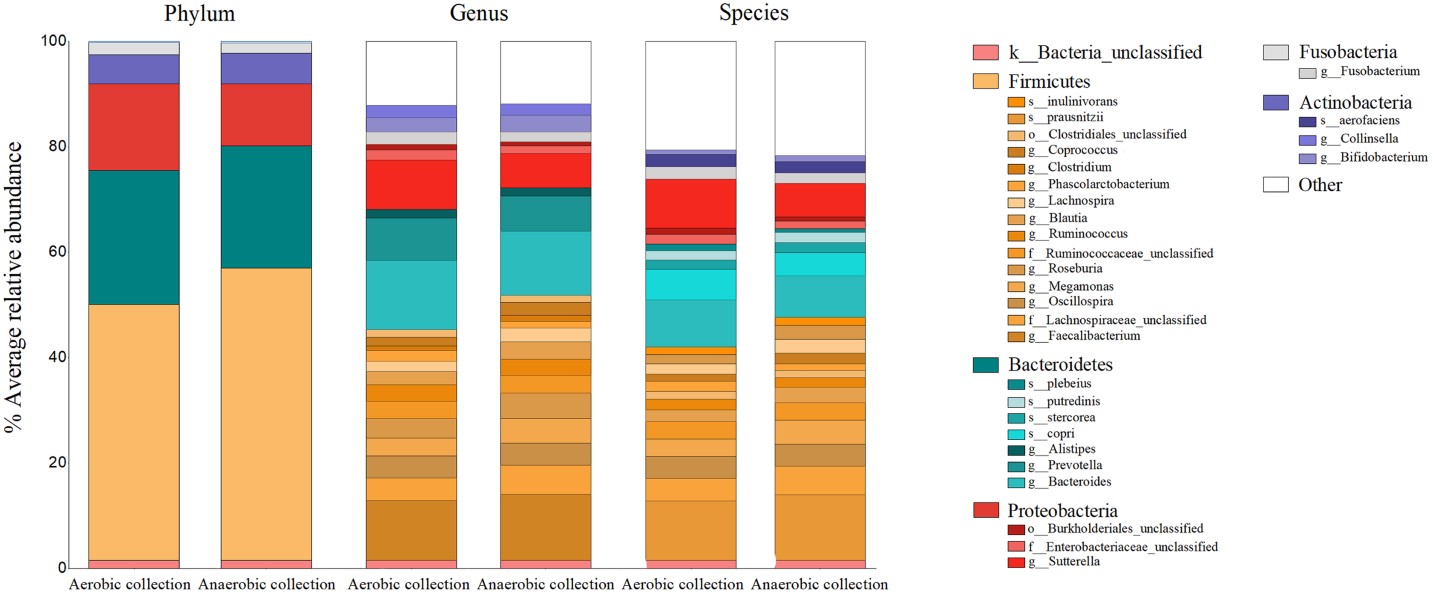

**Figure 2 Relative percent gut microbiota compositions of aerobic and anaerobic transport groups at phylum, genus and species levels.** Color shades represent bacterial phyla: yellow (Firmicutes), blue (Bacteroidetes), red (Proteobacteria), grey (Fusobacteria), purple (Actinobacteria), white (Other), and pink (unclassified bacterial phyla). The OTUs where Mothur could not identify the genus (or species) names were denoted by small letters (o_ abbreviates order; f_, family; g_, genus and s_, species) to the deepest taxonomic names that could be identified; k_ abbreviated kingdom bacteria but unclassified phylum; and "Other" represented <1% phylum (or genus, or species) OTUs. In right-hand legend the names of OTUs were listed from top-to-bottom the same order as in the barchart OTUs (gray lines in barchart to separate OTU names in each phylum).

## Quantitative microbiota composition analyses between aerobic and anaerobic sample transport groups

Following the quantification of bacteria by the universal 16S rRNA gene qPCR, the number of bacterial counts and the quantitative microbiota compositions could be analyzed. The quantity of bacterial counts was not significantly different between aerobic and anaerobic sample transport groups, although slightly lower for the aerobic sample transport group (Fig. 3A: $P = 0.057$). Noted that the relatively low in the aerobic sample transport group was due to ID3a and the relatively high in the anaerobic sample transport group was due to ID1an; if except these two, the average bacterial counts of both groups will even become closer to each other and $P$ value increases (Fig. S2).

**A**

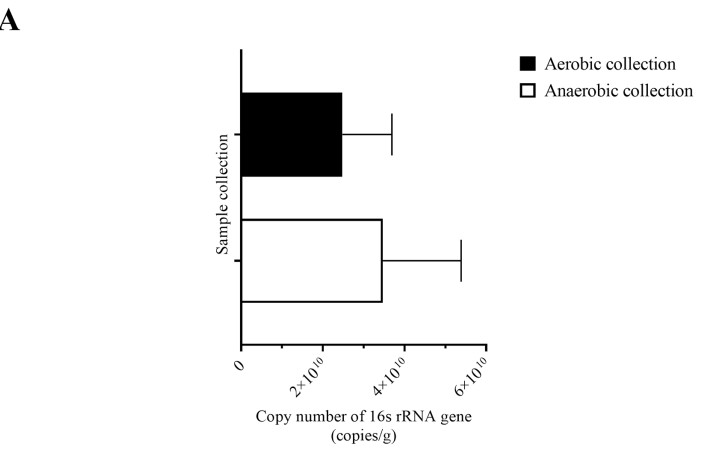

**B**

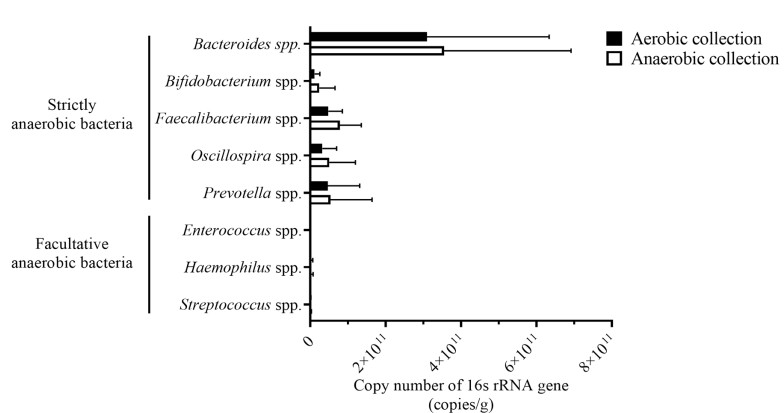

**Figure 3** **Quantification of bacterial counts for (A) average total bacterial counts and (B) average strictly anaerobic and facultative anaerobic bacterial genera, comparing between aerobic and anaerobic sample transport groups.** Data were presented as average ± SD. Statistical differences between groups were tested using Student's *t*-test ($P < 0.05$), and no statistical difference was found.

Next, individual bacterial species corresponding to obligate (or strictly) anaerobes that consisted of five bacterial species and facultative anaerobes that consisted of three species were quantitatively compared. No statistically significant difference in quantity was pointed in these bacterial species between aerobic and anaerobic sample transport groups (Fig. 3B). In detail, the obligate anaerobic *Bacteroides* spp. were found most dominated than other obligate anaerobic bacterial genera in both groups and presented in approximately comparable counts, followed by *Prevotella, Faecalibacterium, Oscillospira, Bifidobacterium,* and the facultative anaerobic *Haemophilus, Streptococcus* and *Enterococcus*, respectively. Nonetheless, the slight but non-statistically significant higher counts of obligate anaerobic bacteria were shown. Still, this trend was minute and found inconsistent for facultative anaerobic bacteria genera (Fig. 3B), highlighting the differences in obligate *vs.* facultative oxygen requirement effect yet at the non-significant statistic. Overall, the percent microbiota composition and the quantitative microbiota did not demonstrate significant difference between aerobic and anaerobic sample transport

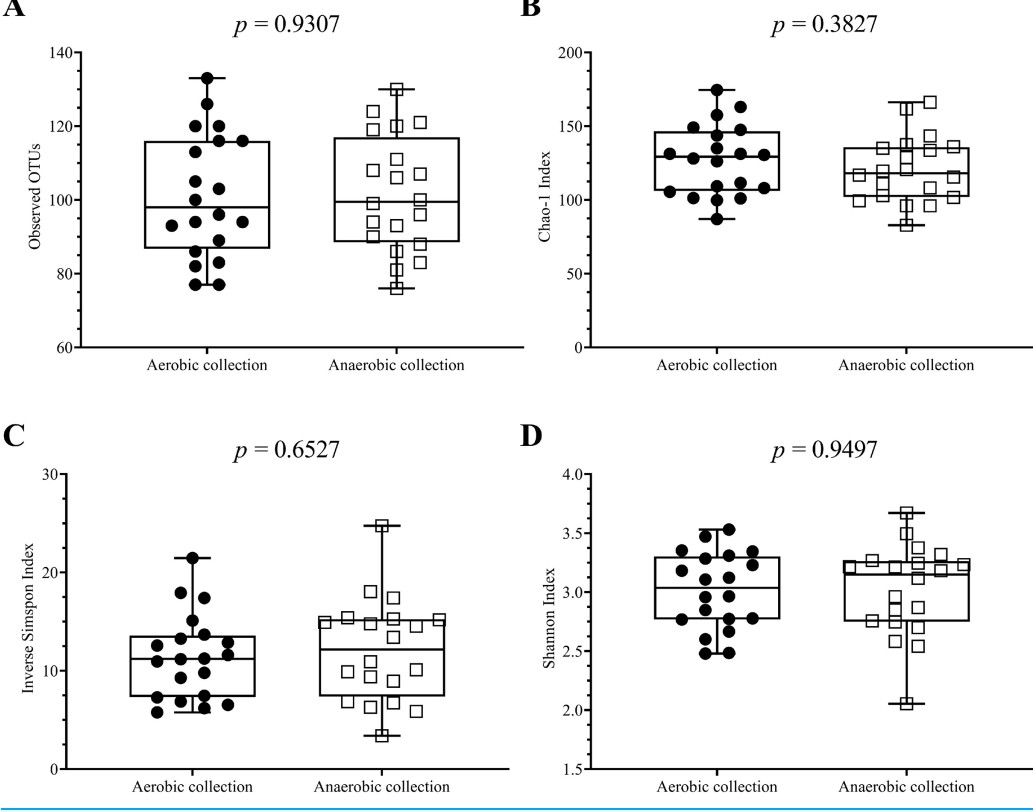

**Figure 4 Scatter plots showing individual and mean range alpha diversity data at species OTUs of aerobic (filled circle) and anaerobic (empty square) sample transport groups, measured by (A) number of OTUs, (B) Chao1 richness, (C) inverse Simpson diversity, and (D).** Statistical differences between groups were tested using Student's $t$-test ($P < 0.05$), and no statistical difference was found: $P > 0.05$.

materials. Subsequently, the alpha diversity by OTU species richness (OTUs and Chao1) and OTU species diversity (inverse Simpson and Shannon) showed very high $P$ values between 0.3827 and 0.9497 (Fig. 4), and the beta diversity among individual samples belonging to aerobic and anaerobic sample transport groups showed no separate clustering pattern (Fig. 5A). Noted that the detail analyses of alpha diversity at OTU phylum and genus levels were also analyzed. No statistic differences were found ($P > 0.05$) (Fig. S3). Additionally, other beta diversity coefficients, such as Sorabund, Morisita-Horn, Thetayc and Bray-Curtis, were computed and all dissimilarity coefficient indices did not separate the microbiota community differences between aerobic and anaerobic sample transport groups (Table S3: $P > 0.05$). Meanwhile, we further classified the samples into healthy and unhealthy categories, and the alpha diversities showed relatively no difference between aerobic and anaerobic sample transports (Fig. S3E).

## Quantitative microbiota analyses between healthy and fat-metabolic disorder groups

When we analyzed the quantitative microbiota structure differences by different beta diversity coefficients, we found the statistical difference between healthy *vs*. fat-metabolic

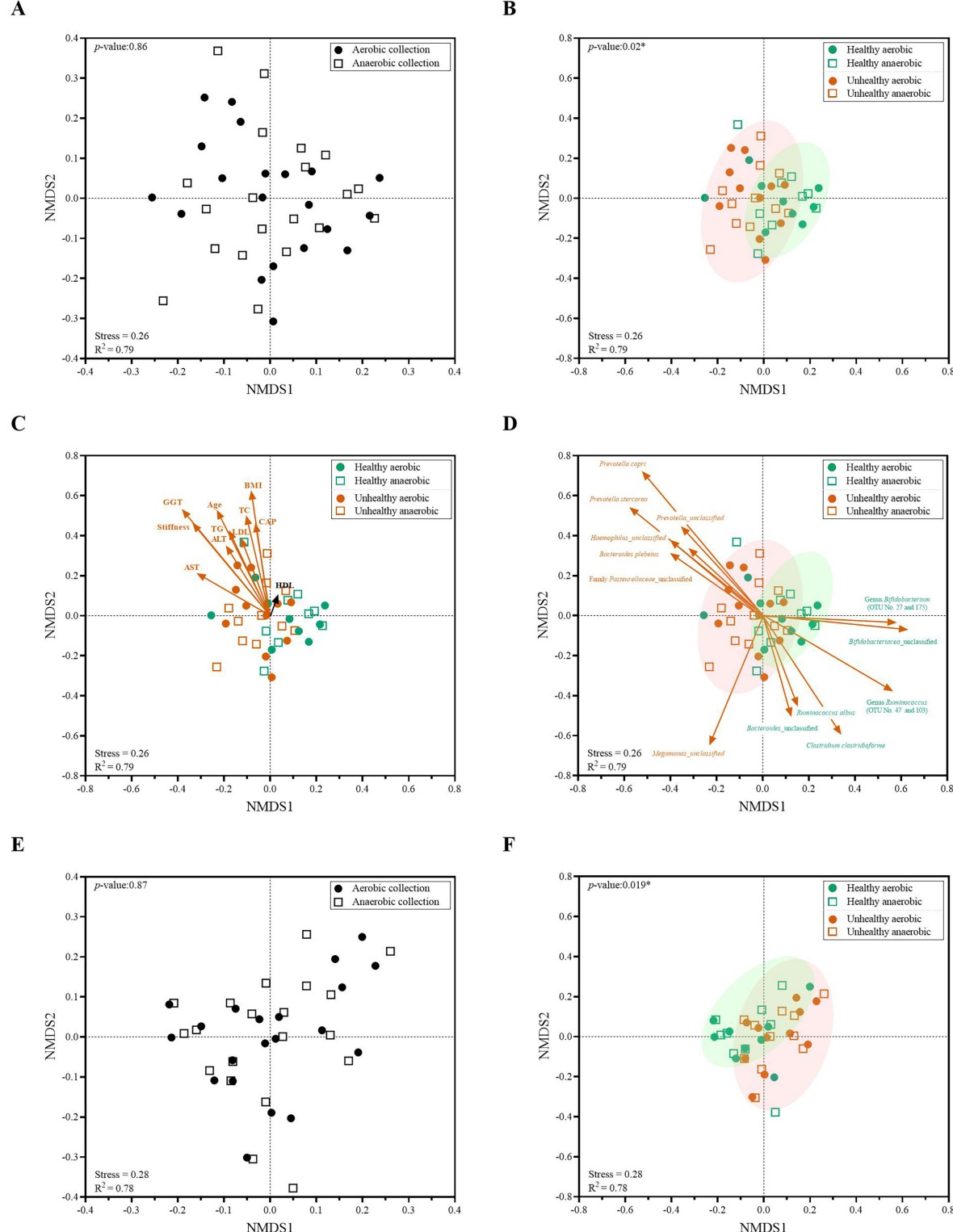

**Figure 5 Non-metric multidimensional scaling (NMDS) constructed from Thetan coefficients displaying beta diversity among quantitative microbiota communities in aspects of (A and E) aerobic and anaerobic sample transport groups and (B–D and F) health and fat-metabolic disorder (denoted "unhealthy") groups.** In (A, B, E and F), AMOVA test was performed to determine statistical separation between designated groups ($P < 0.05$). In (C and D) showed the Pearson's correlations with health status parameters and the representing bacterial species OTUs,
**Figure 5 (continued)**
respectively. A vector direction and length represented the direction and strength of that parameter or OTU to the communities. A red arrow indicated a statistically significant correlation parameter ($P < 0.05$), and a black arrow indicated non-statistically significant correlation parameter ($P > 0.05$). In (C), GGT abbreviates gamma-glutamyl transferase; BMI, body mass index; stiffness, liver stiffness indicates the non-elasticity of the liver associated fat accumulation; TC, total cholesterol; AST, aspartate aminotransferase; ALT, alanine aminotransferase; TG, triglyceride; LDL, low-density lipoproteins; CAP, controlled attenuation parameter; and HDL, high-density lipoproteins. In (E and F), the low-abundance OTUs of <1% or non-relevant inter-individual microbiota were filtered out (remaining as "core microbiota") for the NMDS analysis.

disorder (from now on referred as "unhealthy") groups (Fig. 5B: $P = 0.02$). The differences were found when considering only aerobic healthy *vs.* unhealthy, anaerobic healthy *vs.* unhealthy, and combined aerobic+anaerobic healthy *vs.* unhealthy. Supportively, the clinical parameters corresponding to fat-metabolic disorders demonstrated statistically ($P < 0.05$: age, liver stiffness, GGT, BMI, TC, AST, ALT, TG, LDL, and CAP) and non-statistically ($P > 0.05$, HDL) associated the same direction with the unhealthy microbiota community structure (Fig. 5C). Figure 5D exhibited bacterial species that significantly associated with unhealthy community structure patterns such as *Prevotella*, *Haemophilus* and *Bacteroides plebeius*; and healthy community structure such as *Bifidobacterium*, *Ruminococcus* and *Clostridium*.

Furthermore, the low-abundance OTUs of <1% and non-shared inter-individual microbiota were tested and filtered out (remaining as "core microbiota") for the NMDS analysis. The result remained consistent, demonstrating no statistical difference in quantitative core microbiota between aerobic and anaerobic sample transport groups (Fig. 5E: $P = 0.87$), yet the statistical difference between healthy and unhealthy groups (Fig. 5F: $P = 0.019$). This finding might infer the importance in the core microbiota pattern that aligned the unhealthy microbiota association with the fat-metabolic disorder.

## Metabolic function prediction levels *via* quantitative profiles of pre-valent health-associated bacteria, and microbial metabolic function species biomarkers for healthy and fat-metabolic disorder groups

The metabolic potentials of the potentially important bacteria were analyzed. These included *Bacteroides, Prevotella, Megamonas, Bifidobacterium, Hemophilus, Clostridium, Ruminococcus* and Pasteurellaceae (*Wu, Bushmanc & Lewis, 2013*; *Schirmer et al., 2019*; *Sun et al., 2020*; *Sabo & Dumitrascu, 2021*). The generally most active microbial-related functions were metabolism pathway (49.92%: primarily amino acid and carbohydrate metabolisms followed by energy, cofactors and vitamins, lipid and xenobiotics biodegradation metabolisms), 19.94% in genetic information processing, 16.22% in environmental information processing, 3.11% cellular process, 0.91% human diseases, 0.65% organismal systems, and 5.09% poorly characterized. The OTUs of *Bacteroides* and *Prevotella copri* represented the topmost varying functional metabolisms (Fig. 6A). Meanwhile the functional redundancy among bacterial OTUs, the relative abundances of these health-associated bacteria showed the dynamic functions with some distinguished categories of metabolisms, cellular process, and genetic information processing between healthy and fat-metabolic disorder groups. For instance, the relatively more abundance of

**A**

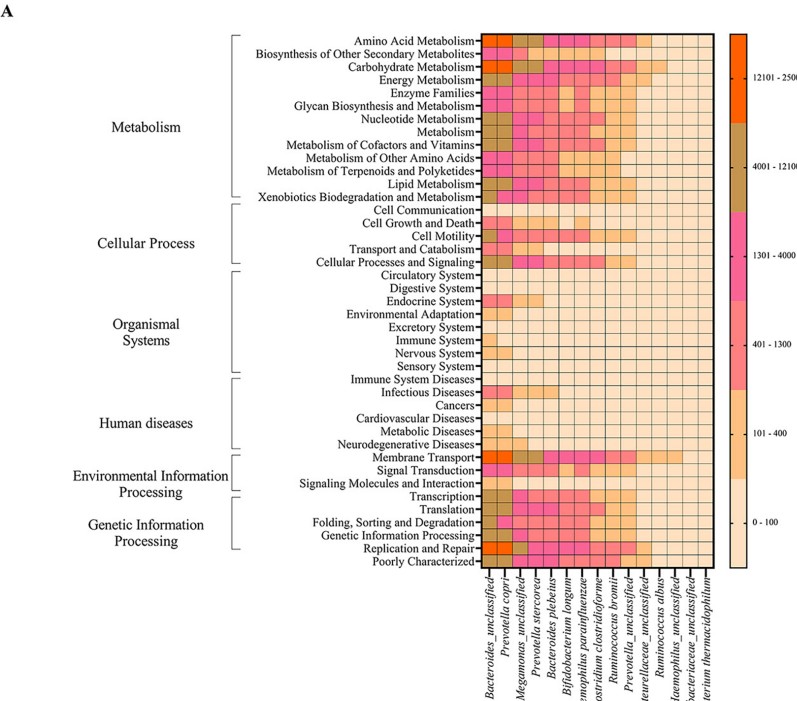

**B**

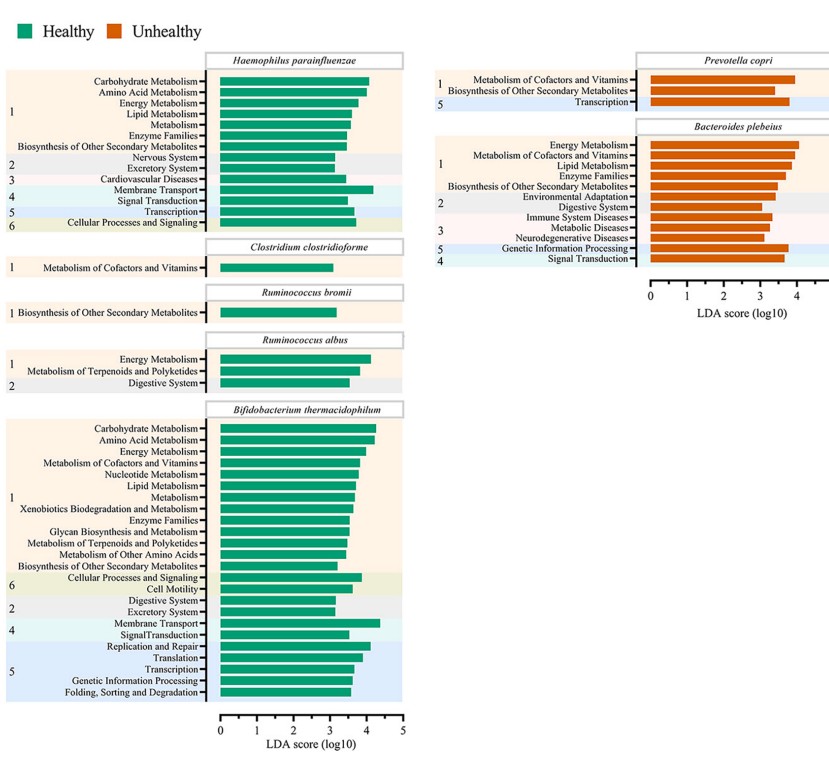

**Figure 6 Metabolic functional prediction associated to quantitative profiles of (A) prevalent health-associated bacteria OTUs and (B) linear discriminant analysis (LDA) combined effect size (LEfSe) as bacterial species and associated microbial metabolic function biomarkers for healthy or fat-metabolic disorder (denoted "unhealthy") groups.** Microbial metabolic functions were estimated

**Figure 6** (continued)
according to KEGG pathways. In (A), a different color from nude to tangerine represents the level of quantitative microbial metabolic function abundance from absence to the highest presence level (scale in vertical bar chart). In (B), a numeric in front of KEGG name represents the KEGG pathway category: 1, metabolism; 2, organismal systems; 3, diseases; 4, environmental information processing; 5, genetic information processing; and 6, cellular processes. The LDA score >3.0 was referred microbial metabolic function markers (ANOVA Welch's test, $P < 0.05$).

amino acids, carbohydrate and energy metabolism functions, cellular processes, genetic information processing, and human diseases were reported in the fat-metabolic disorder group. *Prevotella copri, Prevotella stercorea*, and *Bacteroides plebeius*, were estimated to have more diverse and abundant functions in the fat-metabolic disorder group, while Bacteroides and *Bifidobacterium longum*, were estimated to be more diverse and abundant in the healthy group (Fig. S4). These microbial metabolism differences between groups allowed LEfSe to identify the specific microbial metabolic functions along the bacterial species as the biomarkers to differentiate between healthy *vs*. fat-metabolic disorder groups, with statistical *P* values. *Prevotella copri* and *Bacteroides plebeius* were the biomarkers for the fat-metabolic disorder. Their microbial metabolic functions included many functions involved in diseases (immune system diseases, metabolic diseases, and neurodegenerative diseases). In contrast, the healthy group showed a greater variety of bacterial species and their associated metabolic functions when compared to the unhealthy group. This supports the existence of diverse microbial-related metabolic functions in healthy human guts. It was noted that the commonly reported functions were related to metabolism and organismal systems pathways, meanwhile the human disease pathway was rare in the healthy group (Fig. 6B).

## DISCUSSION

As intestine occupies the most number and diversity of bacteria in human body, fecal (gut) microbiome represents the important field to study bacterial interactions with human heath (or diseases). The fat-metabolic disorder represents one common related disorder with fecal microbiota dysbiosis. Due to variation in sample transport materials, especially in local and limited research resource settings, the anaerobic sample transport materials might be utilized. Hence, this study analyzed influences of aerobic and anaerobic sample transport materials on percent composition and quantitative composition of gut microbiota, and also identified whether these influences could affect the interpretation in microbiomes of healthy compared with the fat-metabolic disorder. Further, we could describe the percent and quantitative microbiota differences (including the core microbiota analyses) in heathy and fat-metabolic disorder subjects disrespect of aerobic or anaerobic sample transport materials.

Our study successfully obtained microbiota results in percent and quantitative compositions. The number of quality sequences in each sample allowed reliable Good's coverage index score for OTU diversity and rarefaction curve. Comparing the entire microbiota diversity changes (and the core microbiota diversity changes) between the

aerobic and the anaerobic sample transport materials, in both percentages and quantitative counts, showed no significant difference. Recently, rare species are increasingly recognized to sometimes present an over-proportional role (*Lynch & Neufeld, 2015*; *Jousset et al., 2017*; *Zeng et al., 2022*), our analyses of both the entire microbiota and the core microbiota in this study showed consistent reports with the statistic association was found mainly in the dominant species. No statistical difference in alpha diversity included numbers of OTUs, Chao1 richness, inverse Simpson and Shannon diversity indices, under uncategorized and categorized healthy-unhealthy conditions.

Analyses of obligate anaerobic and facultative anaerobic bacteria were compared and still no statistical difference in these bacterial species between the aerobic and anaerobic sample transport groups. Supportively, the beta diversity analysis by NMDS could not separate bacterial communities of aerobic from anaerobic sample transport groups ($P = 0.86$). Overall, our study indicated no influence between aerobic and anaerobic sample transport materials during sample collection and sample transport (provided that the metagenomic extraction was performed within 2 days) on fecal microbiota and fecal quantitative microbiota. Our results were consistent with *Taguer, Quillier & Maurice (2021)* that short period of oxygen exposure did not affect the nucleic acid content and changes of bacterial microbiota. Moreover, studies reported that the fecal samples for microbiome studies might be kept without any DNA stabilizer reagent at 4 °C for up to 8 weeks and at −20 °C for the longer period (*Choo, Leong & Rogers, 2015*; *Song et al., 2016*). Some obligate anaerobes could partially reduce pressure of aerobic (oxygen) environment by consuming oxygen *via* their bacterial oxidase enzymes (*Baughn & Malamy, 2004*), for examples, a conserved cytochrome *bd* family enzymes in many bacterial species in phyla Firmicutes, Bacteroidetes, Actinomycetes and Proteobacteria. This allowed these obligate bacteria tolerate in the presence of oxygen for several hours (*Borisov et al., 2021*). Yet, when possible, the minimizing oxygen exposure remains the gold standard fecal collection and transport (*Burz et al., 2019*).

Next, we analyzed if these microbiota communities remained associated and able to be distinguished by a fat-metabolic disorder, an example of well-known disease that could be affected by the gut microbiota dysbiosis (*Rothschild et al., 2018*; *Human Microbiome Project (HMP) Consortium, 2012b*; *Zheng, Liwinski & Elinav, 2020*). The beta diversity analyses by NMDS could distinguish the different microbiota community structures between healthy and this disease state, and many clinical factors representing the fat-metabolic disorders (*Dominianni et al., 2015*; *Loo et al., 2017*; *Liu et al., 2019a*; *Xu, Zhu & Qiu, 2019*; *Zheng, Liwinski & Elinav, 2020*) were statistically correlated with the fat-metabolic disorder microbiota subjects (from both aerobic and anaerobic sample transport groups) (*e.g.* age, liver stiffness, GGT, BMI, and TC). In addition, we could identify the bacterial OTUs that statistically associated with the healthy *vs.* fat-metabolic disorder, their microbial metabolic functions, and the potential biomarkers for bacterial species and correlated metabolisms in healthy *vs.* fat-metabolic disorder. For instances, genera such as *Ruminococcus* and *Bifidobacterium* were also reported previously in healthy human gut and provided functions in short chain fatty acid producers, metabolisms of cofactors and vitamins, biosynthesis of secondary metabolites against gut bacterial

pathogens, energy metabolisms, digestive system, and carbohydrate metabolism (*Ze et al., 2012*; *Christopherson et al., 2014*; *Matijašić et al., 2014*). Noted that the presence of *H. parainfluenzae* was reported no negative effect in gut health (*Kosikowska et al., 2016*; *Tanner et al., 2016*). In comparatively, the microbial functions involved human disease were rare found in the healthy than the fat-metabolic disorder groups (Fig. 6B), provided that the microbial functional redundancy was reported in the human gut microbiota in coherence with our analysis that found many shared species-function relationship (Figs. 6A and S4) (*Vieira-Silva et al., 2016*; *Tian et al., 2020*).

For fat-metabolic disorder group, *Prevotella copri* and *Bacteroides plebeius* had been reported as potential gut pathogens for cardiac valve calcification and cardiovascular disease (*Liu et al., 2019b*). However, the prevalence of genus *Prevotella* could be found in healthy gut, and this genus was reported linked with high-fiber diet consumption (*Arumugam et al., 2011*). Hence, the reason that we observed this genus correlated with the fat-metabolic disorder could be biased by the subjects' diets and lifestyles, which we did not have information in the study. Furthermore, limitation in this study included a small number of samples, which could hinder the correlation and bacterial species identification of the microbiota and quantitative microbiota with the fat-metabolic disorder.

Together, the successful utilization in short-term anaerobic sample collection and transport as the genetic preservation method for the 16S rRNA gene profiling through next generation sequencing and qPCR techniques suggested its expanded use to other metagenomic techniques such as shotgun metagenome sequencing and bacterial genome sequencing. This genetic preservation method should also be valid for virome studies (*Gosalbes et al., 2011*; *Bikel et al., 2021*). Nevertheless, we acknowledged possible microbiota diversity changes due to sample transport. For future studies, in addition to the larger sample size for the more significant statistics, one control metagenomic DNA before sample transport (the original fecal sample microbiota) shall be included to confirm no statistical difference between the microbiota in our short-term aerobic transport samples, and the specific analyses of rare species biosphere (*e.g.* mbDenoise) (*Lynch & Neufeld, 2015*; *Jousset et al., 2017*; *Pan, 2021*; *Zeng et al., 2022*). A series of >48 h period of sample collection-transport time shall be included to investigate the possible longer term of sample collection-transport period.

## CONCLUSIONS

The study first analyzed fecal bacterial microbiota and quantitative microbiota, and revealed no influence of anaerobic sample transport material on the microbiota and quantitative microbiota. This indicated that short-term aerobic sample collection and transport does not statistically affect the microbiota analyses, with ≤4 °C sample storage and sample processing within 48 h are required. Our study aimed to showcase the differences in gut microbiota between healthy individuals and those with fat-metabolic disorder. We collected samples using both aerobic and anaerobic transport methods and analyzed the microbiota's quantitative potentials for microbial metabolism and bacterial species biomarkers in Thai adult subjects. Although the gut microbiota dysbiosis factor

that cause this disease exhibited differences in individuals based on factors such as sex, diet patterns, and lifestyles, we were able to identify commonalities across the subjects tested.

## ACKNOWLEDGEMENTS

The authors acknowledged Matanee Palasuk, Piraya Chathanathon, Paweena Ouying and Chitrasak Kullapanich for their technical assistance or advice.

### Funding

The study was supported by the Thailand Science Research and Innovation Fund Chulalongkorn University (CU_FRB65_hea(68)_131_23_61), Thailand Science Research and Innovation (RDG6150124), and the 90th Anniversary of Chulalongkorn University Scholarship, and Multi-Omics for Functional Products in Food, Cosmetics and Animals Research Unit of Chulalongkorn University. The funders had no role in study design, data collection and analysis, decision to publish, or preparation of the manuscript.

### Grant Disclosures

The following grant information was disclosed by the authors:
Thailand Science Research and Innovation Fund Chulalongkorn University: CU_FRB65_hea(68)_131_23_61.
Thailand Science Research and Innovation: RDG6150124.
90th Anniversary of Chulalongkorn University Scholarship.
Multi-Omics for Functional Products in Food, Cosmetics and Animals Research Unit of Chulalongkorn University.

### Competing Interests

The authors declare that they have no competing interests.

### Author Contributions

- Naruemon Tunsakul performed the experiments, analyzed the data, prepared figures and/or tables, authored or reviewed drafts of the article, and approved the final draft.
- Lampet Wongsaroj performed the experiments, analyzed the data, prepared figures and/or tables, and approved the final draft.
- Kantima Janchot performed the experiments, prepared figures and/or tables, and approved the final draft.
- Krit Pongpirul conceived and designed the experiments, prepared figures and/or tables, and approved the final draft.
- Naraporn Somboonna conceived and designed the experiments, performed the experiments, analyzed the data, prepared figures and/or tables, authored or reviewed drafts of the article, and approved the final draft.

## Human Ethics

The following information was supplied relating to ethical approvals (*i.e.*, approving body and any reference numbers):

Institutional Review Board, Faculty of Medicine, Chulalongkorn University

## Data Availability

The nucleic acid sequences in this study are available in the NCBI Sequence Read Archive database: PRJNA1020208.

## Supplemental Information

Supplemental information for this article can be found online at http://dx.doi.org/10.7717/peerj.17270#supplemental-information.

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
