# Peer review of "Non-significant influence between aerobic and anaerobic sample transport materials on gut (fecal) microbiota in healthy and fat-metabolic disorder Thai adults"

_PeerJ, doi:10.7717/peerj.17270_

## Round 0.1 · original submission · Major Revisions

Please revise your manuscript according to the reviewers' suggestions and provide a detailed rebuttal to each reviewer's question.

Reviewer 1 ·

Basic reporting

The manuscript explores the impact of aerobic and anaerobic sample transport on the gut microbiota, focusing on both healthy Tai adults and those with fat metabolic disorders.

The authors conducted two main experiments, employing 16S rRNA sequencing and 16S rRNA gene qPCR. The results provide valuable insights for fecal microbiota studies. To enhance the manuscript's quality, I recommend addressing the following points:

General Recommendations:
* The study is robust, but the language requires clarification for improved readability. I suggest a thorough review for typos and misspellings and consider engaging a writing coach or copy editor.
* Ensure code availability for replicability. The provided command file for analysis needs restructuring to enhance readability.
* Clearly define parameters and cutoffs for the Mothur pipeline, PICRUSt, and LEfSe analyses. This information is crucial for result replication.
* Is not clear how many samples you consider for the 16S profiling and qPCR analyses, please elaborate on this.
* Verify Figure 5 for discrepancies in the number of samples between Healthy and Unhealthy categories, rectifying any errors in metadata and updating the analysis accordingly.
* It would be interesting to contrast the taxonomy, alfa, and beta diversities between Healthy and Unhealthy individuals (for example, healthy aerobic vs healthy anaerobic, unhealthy aerobic vs unhealthy anaerobic, etc.).

Experimental design

* Specify the database used in the Mothur pipeline (Silva or Greengenes) and clarify the parameters considered during analysis.
* Elaborate on the method combining 16S rRNA sequencing and qPCR data described in lines 187-189 for improved clarity.
* Address inconsistencies in the description of taxonomical groups (for example in lines 234-236) through a thorough rewrite.
* Provide detailed information on the LEfSe analysis, including parameters considered and the approach employed (e.g., one analysis per taxonomical level).
* Is not clear how many samples you consider for the 16S profiling and qPCR analyses, please elaborate on this.

Validity of the findings

Results and Conclusions:
* Consider filtering out low-abundance OTUs to enhance the NMDS analysis's meaningfulness and reduce non-relevant inter-individual differences. If you did, please clarify in the manuscript.
* Discuss preservation methods for metagenomics, metatranscriptomics, or viromics if the analysis was expanded beyond the 16S rRNA gene.
* Acknowledge potential microbiota diversity changes before DNA extraction due to sample transport. Suggest measures to address this issue beyond the mentioned alcohol or RNALater methods.

Additional comments

Figure Feedback:
* Enhance Figure 1's clarity by providing more details on subsections, particularly Sequencing and Bioinformatics, samples/groups, and analysis.
* Address potential color-related readability issues in Figure 2 by selecting a more accessible color palette.
* Ensure completeness of species names in Figure 2, such as "s__aerofaciens."
* Add labels to identify samples in Module A of Figure 5 for improved clarity. Also, explain the number of samples.
* Clarify the disparity in the number of Healthy and Unhealthy samples in Modules B-D of Figure 5, explaining the methods.

·

Basic reporting

This manuscript deals with practical research questions related to the changes in the samples from gut microbiota (oxic and anoxic) conditions based on that. The authors prepare several experiments with samples from health patients and disease patients.
The manuscript is unambiguous. It also has a professional article structure, and the authors share their raw data through a public repository (SRA from NCBI).

However, the authors did not correctly discuss their results, so they must improve this section before its publication at peerJ.

Experimental design

The authors have a research question that is well defined, relevant, and meaningful. Almost all of their methods are described with sufficient detail and information to replicate.

Validity of the findings

Almost all of their analysis is robust, statistically sound, and controlled. However, their conclusions can be improved. Please review the detailed comments and suggestions in the attached file.

Additional comments

Please revise my detailed comments and suggestions in the attached file

Reviewer 3 ·

Basic reporting

There are issues with sentence structure; flow of language and spelling mistakes, which can be improved. Other than that, the introduction lays out the scope of the study, objective and background.

For example: the first sentence:
"Human gastrointestinal microbiota is the most complex and dynamic microbial diversity of estimated trillion bacterial cells, which include culture-independent and anaerobic bacteria." Microbiota is not diversity, it includes trillions of bacterial cells, and encompasses both culture-independent and anaerobic bacteria.

Experimental design

Most of the 16S human microbiome studies employ the V4 primers. Was this tested? At least a line explaining the reason of choosing V3-V5 primers would be great.

Particularly with Miseq600, reads can have upto 300bp in length. With the amplicon size of around 550 base-pair, there is very little overlap during contig generation when we follow MothurSOP.

Validity of the findings

Impact would have been better if the analysis was performed on common primers.

Additional comments

No comment.

---

## Round 0.2 · Minor Revisions

Please attend to all the suggestions made by the reviewers; attending to those indications and profound proofreading will ensure fast manuscript processing.

**Language Note:** The Academic Editor has identified that the English language must be improved. PeerJ can provide language editing services - please contact us at copyediting@peerj.com for pricing (be sure to provide your manuscript number and title). Alternatively, you should make your own arrangements to improve the language quality and provide details in your response letter. – PeerJ Staff

Reviewer 1 ·

Basic reporting

I appreciate the authors' thorough attention to the suggestions and corrections provided by the reviewers in this revised version of the manuscript. The enhancements implemented have significantly elevated the overall quality of the article, and I commend the authors for their dedication. However, to ensure the successful publication of this manuscript, it is imperative to address the following minor suggestions:

* Figure Presentation: In Figure 5, the species names remain incomplete (e.g., '
* Methods: Please clarify the database used in the manuscript (Silva or Greengenes) and specify the parameters considered during the analysis. While the provided code mentions only Silva, if it was the sole database utilized, kindly correct the manuscript
* Spelling and Readability: While this manuscript has improved in readability compared to its previous version, some misspellings still require attention. For example, 'lupus erythrematosous' in line 88 should be corrected to 'lupus erythematosus', 'might affect when interpretation in healthy and gut disease' should be revised to 'might affect interpretation in healthy and gut disease’, and there is a misspelling (utlized) in line 105. I encourage the authors to review the text for further improvements in readability thoroughly.

These minor revisions should be addressed for the final version of the manuscript to ensure its quality and readiness for publication in PeerJ.

Experimental design

No comment

Validity of the findings

No comment

Additional comments

No comment

·

Basic reporting

The authors have adressed well all comments and suggestions. They just need to make a proofread of the all manuscript

Experimental design

The authors have adressed well all comments and suggestions. They just need to make a proofread of the all manuscript

Validity of the findings

The authors have adressed well all comments and suggestions. They just need to make a proofread of the all manuscript

Additional comments

The authors have adressed well all comments and suggestions. They just need to make a proofread of the all manuscript

---

## Round 0.3 · Minor Revisions

Although the reviewer reports that the last revision addresses all the technical issues, I propose some editorial modifications that may improve the quality and readability of the manuscript. Please review them at your convenience and include them in a revised version.

Reviewer 1 ·

Basic reporting

The authors have skillfully addressed the reviewers' suggestions, resulting in a significantly more readable and accessible manuscript. This enhanced clarity will allow readers to better understand and appreciate the importance of the authors' findings.

Experimental design

The paper's research question is well-defined and expertly supported by sound experimental design, methodology, and the appropriate use of bioinformatics and statistical analysis.

Validity of the findings

The authors' concise and well-reasoned conclusions, grounded in their findings, offer a valuable contribution to the field and stimulate further scientific inquiry.

---

## Round 0.4 · accepted · Accept

Thanks for addressing the revisions requested. Now, your manuscript is accepted in PeerJ.